# Everyone's Preference Changes Differently: Weighted Multi-Interest Retrieval Model

## Abstract

User embeddings (vectorized representations of a user) are essential in recommendation systems. Numerous approaches have been proposed to construct a representation for the user in order to find similar items for retrieval tasks, and they have been proven effective in industrial recommendation systems. Recently people have discovered the power of using multiple embeddings to represent a user, with the hope that each embedding represents the user's interest in a certain topic. With multi-interest representation, it's important to model the user's preference over the different topics and how the preference change with time. However, existing approaches either fail to estimate the user's affinity to each interest or unreasonably assume every interest of every user fades with an equal rate with time, thus hurting the performance of candidate retrieval. In this paper, we propose the Multi-Interest Preference (MIP) model, an approach that not only produces multi-interest for users by using the user's sequential engagement more effectively but also automatically learns a set of weights to represent the preference over each embedding so that the candidates can be retrieved from each interest proportionally. Extensive experiments have been done on various industrial-scale datasets to demonstrate the effectiveness of our approach. [1]

## 1 Introduction

Today, the recommendation system is widely used in online platforms to help users discover relevant items and deliver a positive user experience. In the industrial recommendation systems, there are usually billions of entries in the item catalog, which make it impossible to calculate the similarity between a user and every item. The common approach is, illustrated in Figure 1, retrieving only hundreds or thousands of candidate items based on their similarity to the user embedding on an approximate level (e.g. inverted indexes, location sensitive hashing) without consuming too much computational power, and then sending the retrieved candidates to the more nuanced ranking models. Thus, finding effective user embedding is fundamental to the recommendation quality.

The user representations learned from the neural networks are proven to work well on large-scale online platforms, such as Google (Cheng et al., 2016), YouTube (Covington et al., 2016), and Alibaba (Wang et al., 2018). Mostly, the user embeddings are learned by aggregating the item embeddings from the user engagement history, via sequential models (**?**Hidasi et al., 2015; Quadrana et al., 2017; Kang & McAuley, 2018; You et al., 2019). These works usually rely on the sequential model, *e.g.* a recurrent neural network (RNN) model or an attention mechanism, to produce a single embedding that summarizes the user's one or more interests from recent and former actions.

Recently researchers (Epasto & Perozzi, 2019; Weston et al., 2013; Pal et al., 2020; Li et al., 2005) have discovered the importance of having multiple embeddings for an individual, especially in the retrieval phase, with the hope that they can capture a user's multiple interests. The intuition is quite clear: if multiple interests of a user are collapsed into a single embedding, though this embedding could be similar to and can be decoded to all the true interests of the user, directly using the single collapsed embedding to retrieve the closest items might result in items that the user is not quite interested in, as illustrated in Figure 1.

---

[1] The code is available at `https://anonymous.4open.science/r/MIP-802B`

Figure 1: Mis-representation with single user embedding in the retrieve-then-rank framework.

Though, conventional sequential models like RNN or the Transformer network do not naturally produce multiple sequence-level embeddings as desired in the multi-interest user representation. Existing solutions fall into two directions: 1) **split-by-cluster** approaches first cluster the items in the user engagement history by category labels (Li et al., 2019) or item embedding vectors (Pal et al., 2020) and then compute a representation embedding per cluster; 2) **split-by-attention** models adopt transformer-like architecture with two modifications. The query vectors in the attention are learnable vectors instead of the projections from the input and the results of each attention head are directly taken as multiple embeddings (Zhuang et al., 2020; Cen et al., 2020). The limitation of the two approaches is obvious: the split-by-cluster method works best with dense the item feature (Xue et al., 2005); and split-by-attention models bias towards the popular categories owing to its shared query vector among all the users and are inflexible to adjust the number of interests, which is fixed in the training phase as the number of attention heads.

Moreover, the existing multi-interest works ignore one important aspect: the weights for each embedding. In the retrieval stage, given the limited number of items to return, retrieving items from each embedding uniformly will cause a recall problem when the user clearly indicates a high affinity towards one or two categories. Some existing approaches, *e.g.* PinnerSage (Pal et al., 2020), use exponentially decayed weights to assign a higher score to interests that have more frequent and recent engagements. However, the methods still assume that in the same period, regardless of the interest is enduring or ephemeral, the level of interest decays equally for any user. Furthermore, these works also assume the number of embeddings to be fixed across all users. Not only is this hyperparameter costly to find, but also the assumption that all users have the same number of interests is questionable. Some dormant users can be well represented using one or two vectors, while others might have a far more diverse set of niche interests that requires tens of embeddings to represent.

In this paper, we propose Multi-Interest Preference (MIP) model that learns user embeddings on multiple interest dimensions with personalized and context-aware interest weights. The MIP model is consist of a clustering-enhanced multi-head attention module to compute multiple interests and a feed-forward network to predict the weights for each embedding from the interest embedding as well as the temporal patterns of the interest. The clustering-enhanced attention overcomes the aforementioned shortcomings from two aspects: the query, key, and value vectors are projected from the user's engaged items, thus the output of the attention is personalized and minimized the bias toward globally popular categories; moreover, the clustering module can be applied before or after the multi-head attention, releasing the assumption that item features are pre-computed or the item-category labels are available. The main contribution of this paper and the experimental evidence can be summarized as follows:

- We propose a multi-interest user representation model that minimizes the bias towards popular categories and is applicable no matter if the item embeddings are pre-computed. MIP is successful in various industry-scale datasets (Section 4.1, 4.2); Appendix A.1 reveals the bias in global query vector and the error from fixed number of clusters in the split-by-attention approaches, comparing to MIP.

- In addition to the multi-facet vector representations of a user, MIP will assign weights to each embedding, which is automatically customized for each user interest, and improve the recall of candidate generation by retrieving more candidates from the most representative embedding (Section 4.3).

- Although if the cluster algorithms require, MIP still asks for a number of clusters during the training phase, the number of clusters in MIP in the inference phase can be trivially increased

or decreased without re-training of the model. And the experimental results (Appendix A.3) show that re-configuring the number of clusters has an insignificant impact on the retrieval performance, thus allowing the system to trade off the storage and computation cost for better performance. Thus, MIP does not require prior knowledge of the number of interests of users during the model training phase.

## 2 RELATED WORK

This work relates to two important aspects of existing recommendation systems: sequential models and the multi-interest framework.

**Sequential models.** A basic consensus in the recommendation system is that user embeddings should be inferred from the user's historical behavior, and thus the sequential models have been at the heart of recommendation models. A typical and classical sequential model is the Markov Chain (Rendle et al., 2010; He & McAuley, 2016). While Markov Chain captures short-term patterns of engagement sequence well, it fails to make the recommendation that requires memorizing long sequences. With stronger representation power on long sequences, Recurrent neural networks (RNNs), have been adopted for learning user embedding from arbitrarily long sequences, *e.g.* GRU4Rec (Hidasi et al., 2015) and etc. (Xu et al., 2019; Devooght & Bersini, 2017). Besides the standard RNN models, specialized recurrent units are proposed to meet the special need of incorporating certain information, *e.g.* user demographic information (Donkers et al., 2017), global context (Xia et al., 2017), interest drifts with time (Chen et al., 2019), and interaction session (Hidasi et al., 2015). Recently, the success of the Transformer network (Vaswani et al., 2017) has brought evolution to sequential modeling tasks and has been soon adapted to the recommendation models, *e.g.* ComiRec (Cen et al., 2020), BERT4Rec (Sun et al., 2019), TiSASRec (Li et al., 2020), SASRec (Kang & McAuley, 2018), MIND (Li et al., 2019), PinText2 (Zhuang et al., 2020), and also our MIP.

**Multi-interest user representation.** Representing users by multiple embeddings greatly improved the recommendation quality, but not every existing recommendation model can easily extend to a multi-interest framework. Classical collaborative filtering and matrix factorization methods do not naturally produce multiple user embeddings, and so do RNNs and attention-based models. To discover multiple interests from user engagement history, heuristic methods (Jiang et al., 2020; Yue & Xiang, 2012) and unsupervised learning methods like clustering (Pal et al., 2020; Wandabwa et al., 2020) and community mining (Wang, 2007; Yu, 2008) have been adopted. Besides, researchers have made efforts to modify the existing neural networks to produce multiple results, for instance, the capsule network (Li et al., 2019; Sabour et al., 2017; Cen et al., 2020) and multi-head attention models (Li et al., 2020; Cen et al., 2020; Zhou et al., 2018). However, they require an estimation of the number of interests of users as a hyperparameter and do not learn the weight of interests. Therefore, unlike MIP, they produce an equal number of clusters for every user and treat each interest with uniform importance.

**Relation to previous works.** The motivation of MIP is to acquire weighted multiple user embeddings with standard self-attention but without explicit item-category labels. ComiRec and PinText2 use global-query attention to produce multiple embeddings, which introduces a bias toward frequent items or popular categories and the phenomenon is shown in Appendix A.1. Furthermore, they also predefine a number of interests that is uniform for all the users. Tisas and BERT4Rec adopt self-attention but can not learn multiple embeddings. MIND relies on the category labels to produce multiple embeddings from self-attention and capsule networks, while the category labels are sometimes unavailable or vague in other applications, *e.g.* YouTube and Pinterest. PinnerSage produces multiple embeddings without category labels, but requires pre-computed item embeddings. Lastly, MIP learns the weights of the multiple embeddings. The comparison are summarized in Table 1.

## 3 METHODOLOGY

In this section, we formulate the recommendation problem and the neural architecture to model the multiple user interests with preference weights in detail.

| Name | User Embedding | Sequential Model | Additional Input | Preference Weight |
|------|----------------|------------------|------------------|-------------------|
| GRU4Rec | single | RNN | interaction session | N/A |
| TiSASRec | single | time-aware self-attention | timestamps | N/A |
| BERT4Rec | single | self-attention | – | N/A |
| MIND | multiple | label-aware self-attention | category labels | ✗ |
| ComiRec | multiple | global-query attention | – | ✗ |
| PinText2 | multiple | shared global-query attention | – | ✗ |
| PinnerSage | multiple | N/A | – | heuristic |
| MIP | multiple | time-aware SA | timestamps | learned |

Table 1: Comparison of MIP to existing recommendation models.

## 3.1 PROBLEM STATEMENT

Let $\mathcal{I}$ denote the collection of items (with the ID $v$ or the feature vector $\boldsymbol{p}$) in the data store, and the $\mathcal{U}$ represents the set of users. For each of user $u \in \mathcal{U}$, we observe the historical sequence of items that the user has engaged with, $\mathcal{S}^u = (\boldsymbol{p}_{t_1}^u, \boldsymbol{p}_{t_2}^u, ..., \boldsymbol{p}_{t_{|l_u|}}^u)$ or $(v_{t_1}, v_{t_2}, ..., v_{t_{|l_u|}}^u)$, and the timestamps of the engagement $\mathcal{T}^u = (t_1^u, t_2^u, ..., t_{|l_u|}^u)$. The objective is to learn a set of user embedding $\boldsymbol{z}_i^u \in \mathbb{R}^d$ ($i = 1, ..., k$) and their weights $w_i$. Since the user representation is learned only from the history of that user, hereinafter, we omit the superscript for $u$ in both the input and output sides for simplicity. Notations as summarized in Table 2.

We distinguish two cases of the $\mathcal{I}$: **1) Dense feature $\boldsymbol{p}$.** In industrial recommendation systems, the learning of item features is often decoupled from the recommendation model. For instance, the item feature may contain semantic information extracted from natural language models and visionary features from computer vision models. Given those feature vectors, the recommendation object mainly focuses on learning the user embeddings from the history of user behaviors. **2) Sparse feature $v$.** On the contrary, in many public datasets, the items have no attainable feature except item IDs (encoded as one-hot vectors), and the recommendation model must learn the dense representation of items through only the user-item interactions. The proposed model is capable of handling both cases.

## 3.2 MULTI-INTEREST USER REPRESENTATION

**Item Encoding.** We distinguish two conditions of an item. In most practical systems, every item has a pre-computed feature vector (denote as $\boldsymbol{p}_j$) defined in the space where the distance can capture the similarity between the items. In other cases where the recommendation system knows nothing but unique ids (denoted by an one-hot vector $\boldsymbol{v}_j$) about the item, the system needs to learn the dense item feature through an embedding layer:

$$\boldsymbol{p}_j = W_{emb}\boldsymbol{v}_j \tag{1}$$

Additionally, how the engaged items reflect the user's interest may be implied by the sequential order and the time when they interact. Thus, we concatenate the item features with both its positional and

| Notations | Description | Notations | Description |
|-----------|-------------|-----------|-------------|
| $\mathcal{I}, \mathcal{U}$ | Item set and user set | $[;]$ | Vector concatenation operator |
| $l$ | Abbreviation of $l_u$, length of $\mathcal{S}$ | $\mathcal{C}$ | Cluster assignment, $\mathcal{C} \in \mathbb{R}^l$. |
| $d$ | The item embedding dimension | $h$ | Attention head superscript |
| $\boldsymbol{p}$ | An item embedding, $\boldsymbol{p} \in \mathbb{R}^d$ | $W_q^h, \boldsymbol{b}_q^h$ | Query projection weights and bias |
| $\boldsymbol{p}_i, t_i$ | Short form of the $\boldsymbol{p}_{t_i}^u$ and $t_i^u$ | $W_r^h, \boldsymbol{b}_r^h$ | Key projection weights and bias |
| $M$ | Attention mask matrix | $d_{model}$ | Projected key/query vector size |
| $\mathcal{S}$ | Abbreviation of $\mathcal{S}^u$, Engagement history of user $u$ | | |
| $\boldsymbol{z}_i, Z$ | User embedding vector(s), $\boldsymbol{z}_i \in \mathbb{R}^d$, $Z \in \mathbb{R}^{d \times k}$ | | |
| $\mathbb{1}_{[cond]}$ | Indicator function, 1 if $cond$ is true, 0 otherwise | | |
| $k$ | Maximum number of embedding vectors per user | | |

Table 2: Notations

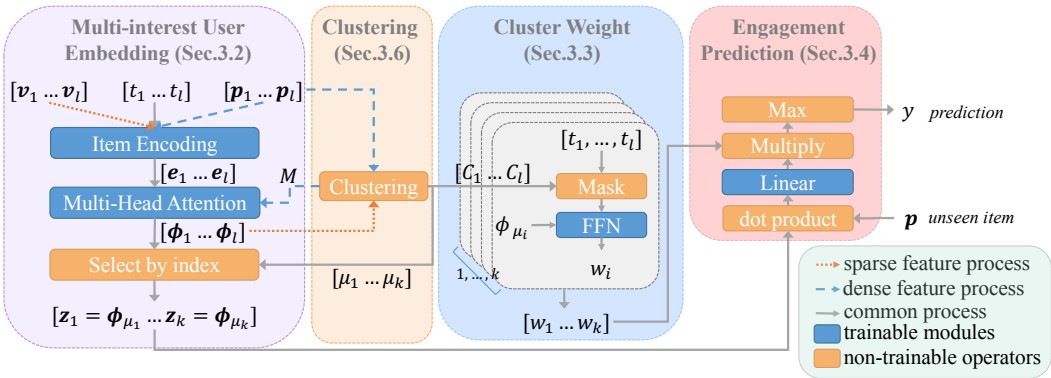

Figure 2: An overview of MIP architecture. The input to the model is the user engagement history containing item embeddings ($[\boldsymbol{v}_1...\boldsymbol{v}_l]$ or $[\boldsymbol{p}_1...\boldsymbol{p}_l]$) and their timestamps ($[t_1...t_l]$). The multi-interest user embedding module produces $k$ embeddings, where $k$ is decided by the clustering method or as a hyperparameter. With clustering and multi-interest representation, the cluster weight module will then estimate the cluster weights for each cluster. Finally, the multi-interest embeddings with corresponding weights are combined to predict the user's interest in an unseen item $\boldsymbol{p}$. The processes with dense and sparse features are shown with different arrows.

temporal information to produce the action encoding:

$$\mathbf{e}_j = [\boldsymbol{p}_j; \boldsymbol{\tau}(t_j); \boldsymbol{\rho}(j)] \tag{2}$$

and the positional ($\boldsymbol{\rho}$) and temporal ($\boldsymbol{\tau}$) encodings are given by Vaswani et al. (2017):

$$\boldsymbol{\tau}_{2j}(t_i) = sin\big(t_i/(\tau_{max})^{2j/m_t}\big), \boldsymbol{\tau}_{2j+1}(t_i) = cos\big(t_i/(\tau_{max})^{2j/m_t}\big)$$
$$\boldsymbol{\rho}_{2j}(i) = sin\big(i/(\rho_{max})^{2j/m_p}\big), \boldsymbol{\rho}_{2j+1}(i) = cos\big(i/(\rho_{max})^{2j/m_p}\big) \tag{3}$$

The hyper-parameters are set as $\tau_{max} = 10^4$, $m_t = 1$, and $m_p = 1$. The unit of timestamps is *day*. In the Section A.2, other forms of encodings are compared, and the Equation 3 has a weak advantage.

**User Representation Candidates** To find only a few vectors to represent the entire user-engaged items, the model has to consider the similarity within the engaged items. The process is achieved through the self-attention layer:

$$s_{i,j}^h = \big((W_q^h \boldsymbol{e}_j + \boldsymbol{b}_q^h)^\top \cdot (W_r^h \boldsymbol{e}_i + \boldsymbol{b}_r^h)\big)/\sqrt{d_{model}}; \quad a_{i,j}^h = softmax_i(s_{i,j}^h) \tag{4}$$

We constrain the attention scores $a$ with a mask matrix $M$. In the sparse feature case, let $M_{i,j} = 1$ for any $i, j$; while in the dense feature case, $M$ indicate the intra-cluster relationship, *i.e.* $M_{i,j} = \mathbb{1}_{[\mathcal{C}_i = \mathcal{C}_j]}$, where the $\mathcal{C}$ denotes the cluster-ID. Thus, $M$ requires the attention model to only aggregate items within the same cluster. The generation of $M$ in the dense feature case is detailed in Section 3.6. With $M$ and attention scores $a$, each attention head aggregates the sequences as:

$$\phi_j^h = \sum_i a_{i,j}^h M_{i,j} \boldsymbol{p}_i \tag{5}$$

To process the aggregated vector from all attention heads, the dropout layer and a feed-forward network (FFN) are applied, and compute the output vectors as

$$\phi_j = FFN(Dropout([\phi_j^1; ...; \phi_j^H])) \tag{6}$$

where $j = 1, ..., l$. The $FFN()$ has two fully-connected layers with a hyperbolic tangent activation function after the first layer, *i.e.* $FFN(x) = W(tanh(W'x + b')) + b$.

Now, each $\phi_j$ summarizes the item features $\boldsymbol{p}_j$, its adjacency with other items, and neighbor item features. The remaining question is which vectors from the $\{\phi_j\}$ are the most representative.

**Cluster Representations** So far, the multi-head attention module has produced $l$ output vectors $\phi_1, ..., \phi_l$, and each $\phi_i$ uses $\boldsymbol{p}_i$ as the (unprojected) query. To find representative vectors, the model utilizes the clustering results. Denote the last item in each cluster as $\boldsymbol{p}_{\mu_1}, ..., \boldsymbol{p}_{\mu_k}$, we take the $\phi_{\mu_*}$ that uses the last item as query, to represent the corresponding clusters. Then let the user embeddings be $\boldsymbol{z}_i = \phi_{\mu_i}$, and let the user representation be $Z \in \mathbb{R}^{k \times d}$, then

$$Z = [\mathbf{z}_1^\top; ...; \mathbf{z}_k^\top] \tag{7}$$

### 3.3 CLUSTER WEIGHT MODULE

Besides the multi-interest assumption, it's also likely that the user favors each interest unequally. As mentioned in the earlier section of this paper, ranking these interests can greatly benefit the candidate generation task given its limited budget. In general, a higher weight should be assigned to an interest cluster if the user engages more with items that belong to it. In order to utilize both the contextual cluster information and the user's engagement sequence in that cluster, we build a two-layer feed-forward network on top of the cluster representation $z_j$ and the temporal encoding of items $\tau$ that belong to the cluster $j$. We will mask those items that belong to other clusters as zero to make the input dimension consistent. The cluster weight can be written as

$$w_j = FFN([z_j; \mathbb{1}_{[C_1 \in L_j]} \cdot \tau_1; ...; \mathbb{1}_{[C_l \in L_j]} \cdot \tau_l]) \tag{8}$$

Concretely, the $FFN()$ consists of two fully-connected layers with a sigmoid function in between, *e.g.* $FFN(x) = W(sigmoid(W'x + b')) + b$. Physically, the clusters learned from Section 3.2 are topics that the user likes, so the preference weights of the clusters should be positive. Therefore, the output of the second layer are normalized to the range of $[0, +\infty]$ by softplus function[2].

### 3.4 USER-ITEM ENGAGEMENT PREDICTION

Intuitively, a user will engage with an item as long as the item match *one* of his/her interests (not *all*). In other words, it's important that the item embedding is very close to one of the user embeddings, rather than being as close as possible to all of them. Therefore, the user-item affinity should depend on the maximum of item embedding and user embedding on each interest dimension. Furthermore, user's affinity to each interest cluster should be embodied in the likelihood as well: a higher weight should be assigned to an interest cluster when the user engages more with items that belong to it. Therefore, we propose the likelihood that a user will engage with an item as follow:

$$y = \max\{w_j Linear(z_j \cdot p)\}_{j=1}^k = \max\{w_j(W_o(z_j \cdot p) + b_o)\}_{j=1}^k \tag{9}$$

where $Z = [z_1^\top; \cdots; z_k^\top]$ is the aforementioned user multi-interest embedding (on $k$ interest dimensions), and $p$ is the item embedding.

### 3.5 LOSS FUNCTION

Given the set of items with a positive label ($\mathcal{I}_+$) and a negative label ($\mathcal{I}_-$), the negative log-likelihood (NLL) loss of our model can be written as:

$$\mathcal{L} = -\sum_{u \in \mathcal{U}} \Big( \sum_{p_i \in \mathcal{I}_+^u} \log(y_i^u) + \sum_{p_i \in \mathcal{I}_-^u} \log(1 - y_i^u) \Big) / \Big( \sum_{u \in \mathcal{U}} (|\mathcal{I}_+^u| + |\mathcal{I}_-^u|) \Big) \tag{10}$$

### 3.6 CLUSTERING

We have referred to a clustering step in producing attention mask ($M$ in Eq. 5), finding cluster representation through last item in each cluster ($\mu_k$ in Eq. 7), and learning the cluster weight ($\mathcal{C}$ in Eq. 8). Generically, the clustering algorithm takes the set of vectors and produces the clusters $L_1, L_2, ..., L_k$ which is a partition of $\mathcal{S}$. From the partition, the following items are computed: cluster assignment $\mathcal{C}$, a list of the same length as $\mathcal{S}$ that $\mathcal{C}_i = r$ if $p_i \in L_r$, $i = 1, ..., l$ and $r = 1, ..., k$; attention mask $M$, a symmetric binary matrix that $M_{i,j} = \mathbb{1}_{[\mathcal{C}_i = \mathcal{C}_j]}$; and last item index $\mu$ that $\mu_r$ denotes the index of the latest item if cluster $r$.

The clustering is applied differently with the dense and sparse features. With dense features, the exogenous item embeddings ($p_j$) are clustered, and the cluster results lead to a non-constant $M$ mask, while with the sparse feature, the clustering is applied to the user representation candidates (Eq. 6). When unspecified, the model uses the Ward Jr (1963)'s clustering method.[3]

|            | P@20  | R@20  | AUC   | NLL   |
|------------|-------|-------|-------|-------|
| PinnerSage | 74.02 | 29.61 | 81.45 | 103.3 |
| TiSASRec   | –     | –     | 84.96 | 47.8  |
| ComiRec    | 86.35 | 34.54 | 87.50 | 40.7  |
| MIP        | **88.20** | **35.28** | **89.26** | **37.7** |

Table 3: Performance on the Pinterest dataset.

|               | Amazon  | MovieLens | Taobao  |
|---------------|---------|-----------|---------|
| # Items       | 425,582 | 15,243    | 823,971 |
| # Interactions| 51M     | 20M       | 100M    |
| # Training    | 57,165  | 127,212   | 343,171 |
| # Test        | 5,000   | 5,000     | 10,000  |
| # Validation  | 5,000   | 5,000     | 10,000  |

Table 4: Dataset statistics.

## 4 EXPERIMENTS

We conduct an exhaustive analysis to demonstrate the effectiveness of MIP on the data from Pinterest, one of the largest online content discovery platforms, and on public datasets.

### 4.1 WITH SPARSE FEATURE

We first evaluate MIP on learning from collaborative filtering datasets, where the item features are absent and will be learned from the user-item interactions.

**Dataset:** Three public datasets are used: Amazon-book [4] (hereinafter, Amazon), Taobao[5], and MovieLens[6]. We adopted a 10-core setting as previous works (Li et al., 2020; Wang et al., 2019) and filtered out rare items that appear less than 10 times in the whole dataset, and the inactive users who interact with less than 100 items. We split each user's engagement history to non-overlapping sequences of length 100, and use the first 50 items to learn the user embedding(s) and the last 50 items as positive samples to rank. For each sequence, another 50 negative samples are uniformly randomly selected from the items that the user does not interact with. The numbers of sequences in the datasets are listed in Table 4.

**Models configurations:** We compare the open-sourced baseline models with MIP. The configurations follow the principles: 1) all the item and user embedding vectors have the same size ($d = 32$); 2) if the models adopt a multi-head attention module, the number of attention heads is the same ($H = 8$); 3) the baseline models should have comparable or more parameters than MIP. We let the hidden size in GRU4Rrec be 128, the key and query projected dimension ($d_{model}$ in Eq. 4) is labeled in place with the results, and if the model contains a position-wise FFN (Eq 6), the FFN is two-layered fully-connected with a hidden size of 32. The BERT4Rec model is originally proposed to predict the item directly as a classification task, which is unrealistic in the retrieval setting, so we take its last BERT output as the user embedding to compute the similarity between user and item, and train with the NLL loss. We disabled the session-parallel mini-batch in GRU4Rec since the session information is absent. We also replace the text encoder in the PinText2 with an item embedding layer since the input in our experiments is items instead of sequences of words.

**Metrics:** The models are evaluated in the retrieval scenario, where the recommendation system needs to recommend a batch of items to the user. The following metrics are used: area under the ROC curve (AUC), negative log-likelihood (NLL), precision at top-K (P@K), and recall at top-K (R@K). The metrics are reported in $10^{-2}$.

**Training:** All the models are trained for 100 epochs on Tesla T4 GPU with an early stop strategy that ceases the training when validation AUC does not improve for 20 epochs. The MIP is trained with two-phase. In the first phase, the parameters in the cluster weight module are frozen, and the weights for any cluster are set as 1. After the model converges in the first phase, the parameter freeze is removed, and then all parameters are trained until converge. Appendix A.4 compared and analyzed this training strategy of MIP.

---

[2]$y = \frac{1}{\beta} log(1 + exp(\beta x))$. We set $\beta = 1$

[3]We adopted the scikit-learn implementation. https://scikit-learn.org/stable/modules/generated/sklearn.cluster.AgglomerativeClustering.html, with n_cluster=5, and other default arguments. In Appendix A.3, other clustering methods are compared.

[4]https://jmcauley.ucsd.edu/data/amazon/

[5]https://tianchi.aliyun.com/dataset/dataDetail?dataId=649

[6]https://www.kaggle.com/grouplens/movielens-20m-dataset

| Model | Params. | Config | | Amazon | | Taobao | | MovieLens | |
|---|---|---|---|---|---|---|---|---|---|
| | | layers | $d_{model}$ | AUC | R@50 | AUC | R@50 | AUC | R@50 |
| GRU4Rec | 66338 | 1 | – | 68.62 | 63.5 | 81.55 | 74.48 | **96.13** | 90.31 |
| BERT4Rec | 50242 | 1 | 64 | 68.11 | 63.15 | 81.47 | 74.52 | 95.95 | 90.11 |
| BERT4Rec | 55426 | 2 | 32 | 72.10 | 66.52 | 81.46 | 74.47 | 96.02 | 90.24 |
| PinText2 | 69634 | 1 | 256 | 55.83 | 54.13 | 71.58 | 66.88 | 88.27 | 81.68 |
| TiSASRec | 67586 | 2 | 64 | 72.11 | 66.67 | 81.46 | 74.43 | 96.02 | 90.16 |
| ComiRec | 67586 | 1 | 256 | 71.72 | 67.36 | 70.92 | 65.61 | 96.25 | 90.65 |
| MIP | 49347 | 1 | 32 | **80.47** | **78.85** | **88.49** | **88.43** | 93.04 | **93.34** |

Table 5: Performance on public datasets. *Params* excludes the parameters in the embedding matrix.

**Results:** The performance is summarized in Table 5. MIP has stronger performance on Amazon and Taobao datasets and is trivially worse than GRU4Rec and ComiRec in AUC on MovieLens. Intuitively, the results might be because the shopping scenario fits more to our multi-interest assumption, users purchase only items from their interested categories, like a type of sports or habits, while the majority of people would watch all types of movies, and if they like the movie largely depends on the movie quality, instead of movie category. In addition, since all models have very close performance, MIP is still a competitive approach in applications that do not support the strong multi-interest assumption.

## 4.2 With Dense Feature

**Dataset:** The dataset contains user engagement history collected from Pinterest, an image-sharing and social media service that allows users to share and discover visual content (images and videos). The interactions between a user and an item (also referred to as a *pin*) are categorized into *impression* (pin is shown to the user), *clickthrough* (user clicks the pin), *re-pin* (user saves the pin into their board collection), and *hide* (user manually hides the pin). In total, there are 38 million interactions from 510 thousand users during three weeks period of time. Each pin is represented as a 256-dimension feature extracted by the PinSage model (Ying et al., 2018).

Active users could have very long engagement sequences in a day. Let the engagement sequence of a user be $\{j_1, j_2, \cdots, j_n\}$, then for each training sample, we use the first $l$ engagements to predict the $l$ future engagements. We also enforce a one-day gap between these two segments, because nearby user engagements are usually very consistent with each other (falling into the same category) and thus make the prediction task easy. $l$ is set to 50, which is the same as the setting in Cen et al. (2020). We treat click-through and re-pin as the positive label and hide (which is less often), and impression (without click or re-pin) as the negative label. Since these negative data have also been recommended to the user at some point, they are likely to be still relevant to the user, and thus correlate with the positive data. In order to alleviate the bias, we introduce the *random* negative data where pins are sampled from the whole set of pins. The entire negative dataset will consist of 50% *observed* negative data (impression and hide), and 50% *random* negative data.

**Baselines and Model Configuration:** We compare the multi-interest models PinnerSage and ComiRec, and the single-embedding model Tisas with the same setting as in Section 4.1.

**Results and Analysis.** As shown in Table 3, MIP outperforms all the state-of-the-art models that are published very recently. The detailed comparison of the model and explanation of the improvement are as follows:

- PinnerSage shares the clustering algorithm with MIP, but differs in 1) the cluster is represented by the medoid, 2) and the cluster weight uses the heuristic model. Instead, MIP learns the cluster representations and weights from data end-to-end from data, which aligns with our intuition.

- TiSASRec has a similar attention model structure as MIP, except only using the last output of the attention model as the user embedding. The comparison confirms the necessity of multi-interest representation, as in ComiRec and MIP.

- Compared to ComiRec, MIP interestingly shows that self-attention has stronger representation power than attention with global-query. In Appendix A.1, we use synthetic data to illustrate the internal difference between the two types of models.

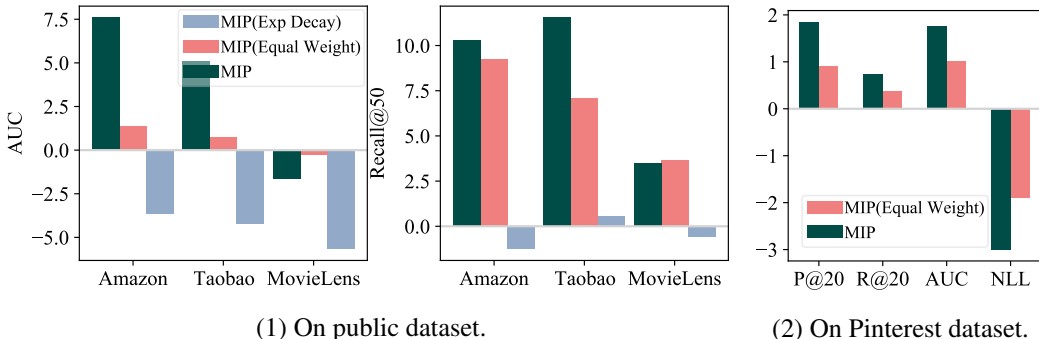

(1) On public dataset.      (2) On Pinterest dataset.

Figure 3: Cluster weight variants performance (difference to the best baseline models)

### 4.3 ABLATION STUDY

This section focus on the claim that the user's preference should be dynamically learned from the temporal pattern by demonstrating the effectiveness of the learned personalized cluster weights module of MIP. We refer the reader to Appendix for a comprehensive ablation study on attention mechanism (A.1), positional and temporal encoding (A.2), re-configuration and comparison of the clustering method post-training (A.3).

**Configurations.** To validate the assumption that the preference trends (weights of multiple interests) change from user to user, we compare the MIP with two variants that disable the cluster weight module. The first one (referred to as *MIP (Equal Weight)*) constantly assigns 1 as cluster weight, thus leading to an equally weighted multi-interest user representation. Another one (referred as *MIP (Exp Decay)*) uses the heuristic formula, an exponential decay weights given by $w_j = \sum_{C_i \in j} exp(-\lambda(t_{now} - t_i))$(Pal et al., 2020). We let the $t_{now}$ be the last user engagement time $t_l$, since in practice it's unrealistic to update the weights in real-time according to the current timestamp. According to Pal et al. (2020), we also set $\lambda = 0.01$, which balances well between emphasizing recently engaged items and accentuating frequently engaged categories. We let the number of clusters $k = 5$.

The variants on the cluster weights are evaluated on Pinterest and the public dataset.

**Results and Analysis** The performances are shown in Figure 3. In most experiments, the AUC and recall can benefit from learned cluster weights (our proposal). From Fig. 3, the equal weights and exponential decay have an advantage on AUC and recall respectively, but learned weights are overall superior to other options. Note that the exponential decay rate is global to any user and any cluster, therefore, the results demonstrated that personalized and context-aware cluster weight estimation is a more reasonable solution to the recommendation.

## 5 CONCLUSIONS

In this paper, we study the problem of multi-interest user embedding for recommendation systems. We follow the recent findings on representing users with multiple embeddings, which has been proven helpful over the single user representation. However, we find that in industrial recommendation systems, it is important to have a set of weights for these multiple embeddings for a more efficient candidate generation process due to its budget on the number of items returned. More specifically, we define the likelihood of an engagement based on the *closest* user embedding to the item embedding and update the weight for the corresponding cluster. We also illustrate the different numbers of interests (embeddings) that users could have, which is fundamentally different from the assumption of similar works. In addition, an attention mechanism is applied in the model architecture, and we have done extensive studies on different model design choices. Finally, case studies on multiple real-world datasets have demonstrated our advantage over state-of-the-art approaches.

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

# A  APPENDIX

## A.1  VARIANTS OF ATTENTION MECHANISM

To interpret the performance improvement of our models against other attention models that have been applied in the recommendation system. We further construct a synthetic dataset and visualize the internal attentions and the user representations aggregated from different attentions.

**Synthetic dataset.** Without loss of generality, we assume there's $G$ global clusters in the corpus, representing different global categories, each of which is a $d$-dimensional Gaussian distribution. Each user is interested in up to $k$ $(< G)$ categories, referred as user clusters. We generate the oracle user interest model by sampling no more than $k$ clusters from $G$ global clusters following a multinomial distribution. Then each of the item in the user engagement history is sampled by two steps: uniformly sample one cluster from user clusters, then sample from the $d$-dimensional Gaussian distribution. Note that in the synthetic model, the item-to-cluster affinity is measured in Euclidean distance, while in the recommendation model, the affinity is decided by cosine distance. To eliminate this discrepancy, we force the Gaussian distributions to center on the unit sphere, so that the rankings by cosine distance and Euclidean distance are consistent.

We use a 2D dataset ($d = 2$) for visualization purpose and another high-dimensional dataset for quantitative evaluation. For 2D dataset, we set a relatively small $G = 8$ and $k = 4$ in order to have a clear boundary between clusters. Since there are only 162 distinct subsets[7] with $G = 8, k = 4$, we use 100 of them for training, 31 for validation and the remaining 31 for testing. For high-dimensional dataset, we set $G = 1024$ and $k = 8$, and let $d = 16, 32, 64, 128$. We generate 10000 users for training, 1000 for validation and another 1000 for testing.

**Attention models.** We focus on comparing our attention model (i.e. *self-attention*), the attention model utilized in ComiRec (i.e. *Non-shared query*), and the one used in PinText2 (i.e. *Shared query*). The comparison of the attentions are in Table 6.

For simplicity, we remove the temporal and positional encoding from the computation of attentions, skip the Ward clustering step from MIP, and directly represent user as Equation 7. Also, the dropout layer is removed in order to eliminate randomness in visualization.

**Metrics.** We visualize the intermediate results and user representations learned from the 2D dataset for qualitative evaluation. For high-dimensional data, we evaluate the performance by AUC and normalized discounted cumulative gain (nDCG).

**Qualitative results and interpretations.** Figure 4 shows the learned user representations given the engagement history. There are three observations. 1) When $H = 1$, global query attention fails to capture all the user interests, while the self attention model is free from the limitation. 2) Viewing from the third row, self-attention model is more accurate in learning cluster representations than global query models. The later is systematically bias due to the global query as shown in global query models Figure 6. 3) All the models learn super-clusters, depending on the bias in the dataset. For the example show in Figure 4, the two adjacent clusters on the top side of the unit circle is often represented to be a super-cluster.

We also visualize the internal attention scores and self attention models (Figure 5). Some attention heads show highly similar attention patterns because their queries are close to each other, which

| Model | Global query | | Self-attention |
|---|---|---|---|
| | Non-shared | Shared | |
| Key vector $\boldsymbol{k}_j$ | $(W_r^h \boldsymbol{p}_j + \boldsymbol{b}^h)$ | | |
| Query vector | $\boldsymbol{q}^h$ | $\boldsymbol{q}$ | $\boldsymbol{q}_i^h = W_q^h \boldsymbol{p}_i + \boldsymbol{b}_q^h$ |
| $\boldsymbol{q}$ shared between sequences | Yes | | No |
| $\boldsymbol{q}$ shared between att. heads | No | Yes | No |
| Dot-product | $e_j^h = \boldsymbol{q}^{h\top} \cdot \boldsymbol{k}_j^h$ | $e_j^h = \boldsymbol{q}^\top \cdot \boldsymbol{k}_j^h$ | $e_{i,j}^h = \boldsymbol{q}_i^{h\top} \cdot \boldsymbol{k}_j^h$ |

Table 6: Variations of multi-head attention.

---

[7]Number of ways to select no more than 4 clusters from a pool of 8 clusters: $162 = \binom{8}{1} + \binom{8}{2} + \binom{8}{3} + \binom{8}{4}$

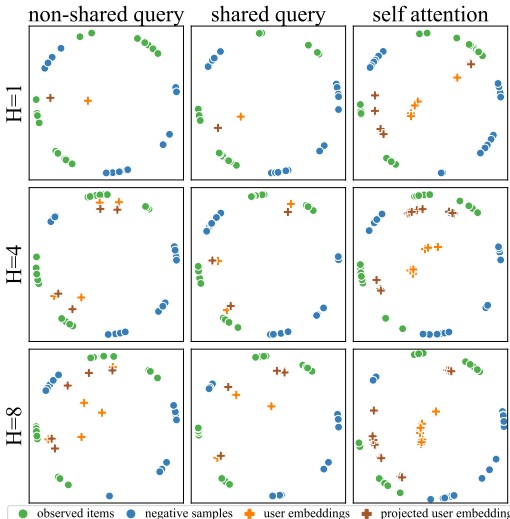

Figure 4: Learned user representations.

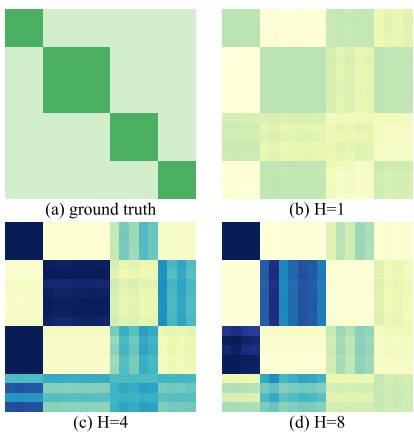

Figure 5: Learned attention scores in self attention model. Darker color represents higher values. (a) the indicator function $\mathbb{1}_{[\mathcal{C}_i=\mathcal{C}_j]}$, (b-d) $a_{i,j}$. The input sequence is re-ordered for better visualization.

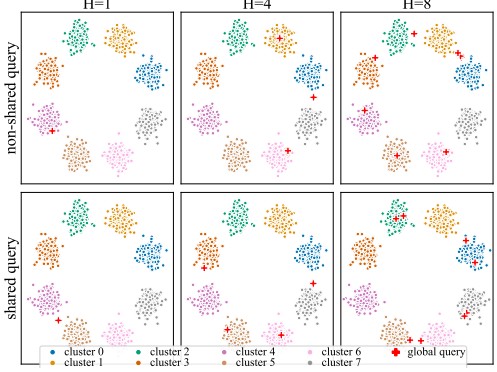

Figure 6: Learned global interests in global query models. The query vector is reversely projected and normalized.

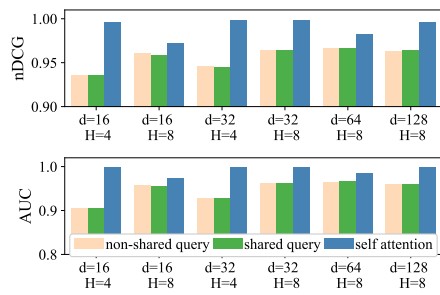

Figure 7: Performance comparison on high dimensional synthetic dataset. $d$ denotes the feature dimension and $H$ is the number of attention heads.

can be verified from Figure 6. Figure 5 compares the ground truth attention model with the learned attention. The learned attention shows clear boundaries between clusters in the heatmap. Note that the ground truth ignores the adjacency of clusters but the self attention model considers the similarity between clusters, so Figure 5(a) is block diagonal while Figure 5(b-d) has dark blocks off the diagonal.

**Quantitative results.** Previous results show intuitive comparison between global query models and the self attention model, and the quantitative results further confirm the consistency of performance gain of self attention. Experiments are repeated on dataset for feature dimension $d = 16, 32, 64, 128$ and number of attention heads $H = 4, 8$. Figure 7 shows that the MIP model constantly and significantly outperforms global query models. As illustrated in the 2D dataset, the performance gain benefits from the personalized user representation, rather than matching to the globally popular clusters. Another observation from the result is that for global query models, $H = 4$ under-performs $H = 8$ models, as the number of attention head decides the number of global clusters the model can learn; however for self attention model, $H = 4$ performs even better than $H = 8$. The explanation is that self-attention model does not require growing number of attention heads with respect to the number of global clusters, and $H = 4$ could be already enough for capturing user interest but easier than $H = 8$ to train.

| Dataset | Configuration | AUC | Recall |
|---------|---------------|-----|--------|
| Amazon | item embedding | 76.34 | 73.75 |
| | +positional | 76.19 | 74.28 |
| | +temporal | 78.59 | 76.94 |
| | +both | **79.31** | **78.22** |
| Taobao | item embedding | 82.06 | 81.75 |
| | +positional | 86.54 | **86.22** |
| | +temporal | **86.72** | 86.56 |
| | +both | 86.59 | 85.83 |
| MovieLens | item embedding | **95.26** | **94.45** |
| | +positional | 95.01 | 94.31 |
| | +temporal | 94.96 | 94.19 |
| | + both | 94.61 | 94.12 |

Table 7: Ablation study on the positional and temporal encoding on public datasets. (in $10^{-2}$)

| Configuration | AUC | NLL |
|---------------|-----|-----|
| item embedding | 0.8923 | **0.377** |
| + positional | 0.8846 | 0.386 |
| + temporal (one-hot) | 0.8850 | 0.388 |
| + temporal (two-hot) | 0.8846 | 0.385 |
| + temporal (sinusoid) | 0.8921 | **0.377** |
| + both (one-hot) | 0.8861 | 0.382 |
| + both (two-hot) | 0.8852 | 0.387 |
| + both (sinusoid) | **0.8926** | **0.377** |

Table 8: Influence of temporal and positional encoding in attention on the performance in MIP

## A.2 POSITIONAL AND TEMPORAL ENCODING

In Eq. 2, the sequential (positional) and temporal information are encoded and included in the self-attention module to produce the multi-interest representations. The motivation is that given items from the same category, the recent ones might better represent the user's current interest than the obsolete items. We verify 1) if the incorporation of positional and temporal encoding is critical to the performance; 2) how the encoding (Eq. 3) method affects the performance.

**Configuration.** The MIP are configured on the two set of choices: the Eq. 2 can be configured alternatively:

- item embedding only: $e_j = p_j$.
- + positional: $e_j = [p_j; \rho(j)]$, where $\rho(j)$ is given by Eq. 3.
- + temporal: $e_j = [p_j; \tau(t_j)]$, where $\tau(t_j)$ is given by Eq. 3.
- + positional and temporal: Eq. 2 and Eq. 3

and there are several other choices of temporal encodings other than the sinusoidal form in the Eq. 3:

- One-hot (Zhou et al., 2018): Create exponential buckets $[0, b), [b, b^2), \cdots, [b^{k-1}, \infty)$ with base $b$, and encode the timestamp as an one-hot vector, i.e. $\tau_i = lookup(buckets(t))$.
- Two-hot (Shi et al., 2021): Similarly create exponential boundaries $\{0, b, b^2, \cdots, b^{k-1}, \infty\}$, and encode the timestamp as $\tau_i = log_b(t) - i$ and $\tau_{i+1} = i + 1 - log_b(t)$, where $b^i \leq t < b^{i+1}$.

**Dataset and Results.** The options to include positional and temporal information are evaluated on all the dataset (Tab. 7), and the encodings methods are compared exhaustively on Pinterest dataset (Tab. 8).

**Analysis.** Two conclusions can be made from Tab. 7 and Tab. 8: 1) including both temporal and positional information is a safe option, which has the best performance on Amazon and Pinterest and marginally ($< 0.01$) worse performance on Taobao and MovieLens; and 2) the model is insensitive to encoding methods.

## A.3 CLUSTERING OPTIONS

The Ward's algorithm is applied to MIP considering its success in PinnerSage(Pal et al., 2020), it's beneficial to explore the selection of clustering algorithm and the number of clusters on the collaborative filtering dataset. To illustrate the impact, we evaluate MIP with a wide range of clustering algorithms.

**Model Configuration and training**: MIP models are configured with an attention module that takes both positional and temporal encoding. For unweighted MIP, no clustering method is applied to the encoded user engagement history $\{z_*\}$ (computed from Eq. 6) in the training stage. For weighted MIP, Ward's algorithm is applied to $\{z_*\}$ and the number of clusters is set to 5. To keep the MIP

| Clustering Method | Inference Clusters | Unweighted MIP | | | Weighted MIP | | |
|---|---|---|---|---|---|---|---|
| | | Amazon | Taobao | MovieLens | Amazon | Taobao | MovieLens |
| None | - | 73.11 | 82.09 | 95.97 | - | - | - |
| Ward | 5 | 71.56 | 80.58 | 95.53 | 79.31 | 86.49 | 94.61 |
| | 8 | 71.99 | 80.99 | 95.72 | 80.47 | 87.85 | 95.25 |
| | 10 | 72.16 | 81.20 | 95.78 | 80.84 | 88.42 | 95.25 |
| K-Means | 5 | 71.58 | 80.62 | 95.53 | 79.26 | 86.18 | 94.86 |
| | 8 | 71.95 | 81.03 | 95.71 | 80.66 | 88.02 | 95.17 |
| | 10 | 72.14 | 81.22 | 95.77 | 80.62 | 88.61 | 95.10 |
| Spectral | 5 | 72.28 | 80.72 | 95.54 | 78.99 | 85.84 | 94.46 |
| | 8 | 72.37 | 81.08 | 95.73 | 80.79 | 87.61 | 94.81 |
| | 10 | 72.64 | 81.26 | 95.78 | 81.19 | 88.40 | 95.07 |
| BIRCH | 5 | 71.98 | 80.63 | 95.52 | 79.39 | 86.29 | 94.61 |
| | 8 | 72.03 | 81.02 | 95.71 | 80.65 | 88.03 | 95.25 |
| | 10 | 72.44 | 81.21 | 95.78 | 80.91 | 88.53 | 95.25 |
| DBSCAN | - | 71.98 | 80.63 | 95.52 | 70.05 | 75.58 | 89.63 |

Table 9: Comparison of clustering options in AUC (in $10^{-2}$). Note that the number of inference clusters is independent of training, i.e. changing the number of inference clusters does not require the re-train of the model.

fully differentiable, the cluster embedding is the encoding of the last item in each cluster, instead of the medoid.

**Inference**: The choice of the clustering in the inference phase is independent of its configuration during the training. We explore the inference options on the pre-trained models. Different types of clustering methods are compared:

- Ward: hierarchical clustering method that minimizes the sum of squared distances within all clusters.
- K-Means: an iterative method also minimized the sum of in-cluster summed squared distances.
- Spectral(Shi & Malik, 2000): performs clustering on the projection of the normalized Laplacian computed from the affinity matrix.
- BIRCH(Zhang et al., 1996): another hierarchical method that clusters the points by building the Clustering Feature Tree.
- DBSCAN(Ester et al., 1996): a density-based clustering method that does not require specifying the number of clusters.

The number of clusters is set to 5, 8, and 10 when required. Note that during training, the number of clusters is fixed to 5, however, after training, MIP can produce other numbers of embeddings per user, which gives the system huge flexibility to trade-off between storage/computation cost and recommendation performance.

**Result and analysis**: There are two observations from Table 9. 1) the different clustering algorithm has a marginal impact on the performance. While PinnerSage reported that Ward's algorithm outperforms the K-Means, their result does not conflict with our observation here. Recall that for PinnerSage and our experiment on the Pinterest dataset, the clustering method is applied to the exogenous item embeddings, thus the clustering methods can be influenced by the non-flat geometry and outliers. However, with the collaborative filtering dataset, the clustering method is applied to the encodings produced by multi-head self-attention layers which average the embedding of the items and all other items (Eq. 4). The encodings after the multi-head self-attention should be smoothly distributed, and as a result, any clustering methods work almost equally well on that. 2) Selecting the number of clusters is a non-trivial trade-off. The motivation to decrease the number of clusters is the storage and computation cost which grow linearly as the number of clusters increases. For unweighted MIP, though the non-clustering (each item is a cluster) settings have the best AUC, decreasing the number of user embedding from 50 (non-clustering) to 10 is still acceptable. For weighted MIP, since it's impossible to learn the clustering weights without applying a clustering method, the trade-off can be more complicated: besides the storage concern, when the number of clusters increases the

| MIP | Amazon | | | Taobao | | | MovieLens | | |
|---|---|---|---|---|---|---|---|---|---|
| Model | Epoch | AUC | R@50 | Epoch | AUC | R@50 | Epoch | AUC | R@50 |
| equal weight | 22 | 73.11 | 66.67 | 4 | 82.09 | 74.86 | 34 | 95.97 | 90.06 |
| two-phase | 6 | 80.47 | 78.85 | 4 | 88.49 | 88.43 | 6 | 93.04 | 93.34 |
| direct-train | 18 | 57.61 | 57.47 | 28 | 80.33 | 80.23 | 14 | 92.43 | 93.68 |

Table 10: Comparison of training strategy and the performance of weighted MIP. The columns **Epoch** shows the training epochs when the best validation AUC is achieved.

| Latency/ Recall | GRU4Rec | BERT4Rec ($L = 1$) | BERT4Rec ($l = 2$) | PinText2 | TiSASRec | ComiRec | MIP (total) | MIP (clustering) |
|---|---|---|---|---|---|---|---|---|
| Train | 218.57 | 925.36 | 1058.34 | 555.79 | 452.94 | 479.59 | 998.62 | 5.86 |
| (std) | (0.81) | (108.96) | (733.33) | (76.96) | (67.07) | (67.50) | (56.31) | (0.76) |
| Inference | 1.15 | 38.34 | 57.53 | 14.46 | 14.54 | 14.61 | 40.05 | 5.93 |
| (std) | (0.20) | (56.60) | (56.23) | (54.99) | (56.34) | (55.70) | (27.08) | (0.72) |
| R@50 | 63.50 | 63.15 | 66.52 | 54.13 | 66.67 | 67.36 | 78.85 | - |

Table 11: Latency and performance comparison of the models. Training and inference latencies are measured in *ms*, and brackets show the standard deviations.

average information to learn the weights of each cluster decreases and consequently may hurt the overall performance; on the other hand, 10-cluster settings are better than the 5-cluster settings for all the dataset.

### A.4 TRAINING WEIGHTED MIP WITH SPARSE FEATURE DATASET

As mentioned in Section 4.1, when training MIP on a collaborative filtering dataset, we first train the equally weighted MIP until converging, then lift the parameter freezing to train the cluster weight module together. We detailed the reason and explanation of the strategy here.

**Configurations.** The model is trained twice with the same configuration in Section 4.1. The first training adopts a two-phase strategy. The first phase model is denoted as *MIP (Equal Weight)*, and the final model is denoted as *MIP (two-phase)*. In the second round, the model is directly trained from random initialization and is denoted as *MIP (direct-train)*.

**Results**: Table 10 shows the performance and number of training epochs to achieve the best performance. The difference between the two-phase and direct-train is significant. Two-phase training not only shortens the training process but also achieves remarkably better results.

**Analysis**: It's unsurprising that two-phase can accelerate the training, but we need to understand why random-initialized MIP under-performs equally weighted MIP, while the MIP is a generalization to equally weighted MIP. The clue is lying in the experiment on the Pinterest dataset. On the Pinterest dataset, the weighted model is randomly initialized and outperforms the equally weighted one. The difference between the two experiments is two-fold: 1) the Pinterest dataset contains dense item features thus no item embedding matrix needs to be learned; 2) the clustering algorithm is applied to the item embedding in the Pinterest dataset but to the output of multi-head self-attention (MHA) in the collaborative filter dataset. As a result, in the Pinterest experiments, the clustering assignment is reasonably reliable throughout the training, since the item embeddings are always meaningful. On the contrary, on the collaborative filtering dataset, with the random initialization, the item encodings after MHA are much noisier at the beginning of the training, and thus the cluster assignment can be fairly random. Then the input and output of the cluster weight module may be meaningless, which causes hardship in learning the weighted MIP.

### A.5 MODEL LATENCY COMPARISON

Seeing the performance gain, another prominent question will be what is the time cost of the performance increase. In this section, we profiled the model latency on a desktop computer with a 12-core Intel i7-8700k CPU, and a single Nvidia GeForce RTX 2080 Ti GPU. The neural network training and inference is on the GPU with vanilla PyTorch framework (version 1.12) without any further optimization on the computation. The clustering algorithm in MIP is performed by CPU with

| NLL | | Triplet ($\alpha$ =0.2) | | Triplet ($\alpha$ =0.5) | | Triplet ($\alpha$ =0.8) | |
|---|---|---|---|---|---|---|---|
| AUC | R@50 | AUC | R@50 | AUC | R@50 | AUC | R@50 |
| 92.32 | 92.97 | 88.46 | 88.20 | 90.33 | 90.08 | 90.60 | 90.61 |

Table 12: MIP Performance on MovieLens with difference loss functions.

Python's scikit-learn package. We set the batch size to 1 and the dataset to Amazon, then measure and summarize the training and inference latency in Table 11.

There are a few observations from Table 11. First, comparing to the neural network inference latency, the clustering step time cost is trivial. PinText2, ComiRec, and TiSASRec has similar training and inference latency, while the performance of them are worse than MIP. BERT4Rec has similar latency as MIP since our sequential model architecture are similar, while the BERT4Rec has worse performance. GRU4Rec has the least inference time. Notice that the standard deviations of the inference latency of PinText2, TiSASRec, and ComiRec are large. It indicates, though on average the three models are faster in inference, MIP inference latency is less possible to be very large while the other latency might be several times longer than average.

Conclusively, MIP, as well as other baselines compared, can all satisfy the latency requirement when applying online, even without further optimization on the computation and serving. MIP has higher time cost compared to some of the baselines, but the performance increase is also appealing.

### A.6 LOSS FUNCTION

Beside the NLL loss function in Equation 10, triplet loss is also widely used in contrastive learning and recommendation system. Let the anchor vector to be $\boldsymbol{p}^a$, the positive vector is $\boldsymbol{p}^+$ and the negative one is $\boldsymbol{p}^-$, the triplet loss is given by:

$$\mathcal{L}_\alpha = \sum_{\mu \in \mathcal{U}} \big( sim(\boldsymbol{p}^a, \boldsymbol{p}^+) - sim(\boldsymbol{p}^a, \boldsymbol{p}^-) + \alpha \big) \tag{11}$$

In the single user embedding models, the $\boldsymbol{p}^a$ is the single user embedding, and $\boldsymbol{p}^+, \boldsymbol{p}^-$ are positive and negative vectors. While in our multiple embedding assumption, to train with the triplet loss, we have to modify the form. Similarly to Equation 9, we introduce the maximum operator in the similarity measure. Define the triplet loss for multiple embedding $\{\boldsymbol{z}_j\}_{j=1}^k$ as:

$$\mathcal{L}_\alpha = \sum_{\mu \in \mathcal{U}} \big( max\{sim(\boldsymbol{z}_j, \boldsymbol{p}^+)\}_{j=1}^k - max\{sim(\boldsymbol{z}_j, \boldsymbol{p}^-)\}_{j=1}^k + \alpha \big) = \sum_{\mu \in \mathcal{U}} (y^+ - y^- + \alpha) \tag{12}$$

where $\alpha$ is a hyperparameter of the positive-negative margin. We let the MIP to train on MovieLens dataset to investigate the impact to the performance of loss function choice. From the observation in Appendix A.4, the MovieLens dataset can be learnt well without two phase training, so we directly train from scratch.

The results in Table 12 illustrates that the triplet loss marginally performs worse than the NLL loss we used. A possible reason might be the migration from single-anchor to muli-anchor, but the further investigation is left as future work.

