# OpenReview forum: "Everyone's Preference Changes Differently: Weighted Multi-Interest Retrieval Model"
_ICLR.cc/2023/Conference — Submitted to ICLR 2023_

### Official Review · Reviewer_QxyP · 2022-10-25

**Confidence:** 3
**Correctness:** 3
**Technical Novelty And Significance:** 3
**Empirical Novelty And Significance:** 3
**Recommendation:** 6

**Clarity, Quality, Novelty And Reproducibility:**

The quality of the work is OK, but the presentation of the paper can be improved.

**Strength And Weaknesses:**

Strength
+ The introduction and its motivation are clear.
+ The performance seems impressive.
+ The ablation tests are reasonable.

Weakness
- The primary issue of this work regards the presentation.
   1) Table 2 misses many notations (e.g., v, \phi, M, etc.)
   2) The notation of the concatenation should be [;]?
   3) Are d and d_model the same?
   4) Why not provide the math equation(s) of FFN() and Linear()?
   5) p is learned from v if it is a cold start data? Are they mutually exclusive inputs?
   6) Does the order of selected \phi matter?


**Summary Of The Paper:**

This work presents a multi-interest retrieval model, the objective of which is to increase the performance of the retrieval stage in a standard two-stage recommendation system. Extensive experiments have been done on various large-scale datasets to show the effectiveness of the proposed approach.

**Summary Of The Review:**

The authors propose an effective recommendation algorithm that produces multi-interest for users and learns a set of weights to represent the preference over each embedding so that the candidates can be retrieved from each interest proportionally. Overall, the paper is readable and the contributions are significant and somewhat new.

---

### Official Review · Reviewer_TbF3 · 2022-10-25

**Confidence:** 3
**Clarity, Quality, Novelty And Reproducibility:** The work is novel, and the quality an…
**Correctness:** 3
**Technical Novelty And Significance:** 3
**Empirical Novelty And Significance:** 3
**Recommendation:** 6

**Strength And Weaknesses:**

S1. The motivation for assigning weights to different user interests is strong and interesting, and the proposed method of deriving importance weight according to user's engagement with each cluster is novel and intuitive.

S2. They conduct extensive experiments on three public datasets. Overall, the improvements of the proposed method are significant and promising.

---

W1. A major concern is the efficiency of MIP. For each instance, they perform clustering on the user behavior sequence, which seems to be very time-consuming during both model training and inference. The time cost of  MIP should be illustrated to show whether it could be applied to real-world recommendation systems.

W2. I also have several questions about the details of MIP. Most of them are about the motivation of designs in the method.

Q1. The paper discriminates two situations of item features: dense features and sparse features. Why is this needed? In real-world systems, it is common that items are related to both sparse and dense features. How can you tackle this problem?

Q2. What is the benefit of performing self-attention on the user behavior sequence before clustering? What if removing the self-attention part?

Q3. Why use the last item in each cluster as the cluster representation?

Q4. Why do you need to input cluster representation along with engagement vector to predict cluster weight in Equation (8)?

Q5. Is the method guaranteed to reach convergence considering it has an inserted clustering module?

---

Minor: What does the number mean in the training/test/validation rows of Table 4?

**Summary Of The Paper:**

This paper proposes a multi-interest user embedding retrieval model MIP for recommendation. The idea is to apply clustering to each user's historical-interacted item embedding sequence. Each cluster is then multiplied with a weight to discriminate their importance.

As they explicitly use clustering methods on item embeddings, the cluster assignment of each item is known. This can be used to derive the importance weights of clusters, i.e., a higher weight should be assigned to an interest cluster if the user engages more with items that belong to it.

They conduct extensive experiments on three public datasets to conclude the superior performance of MIP in contrast to other retrieval models.

**Summary Of The Review:**

Due to the above concerns (mostly due to the efficiency problem in W1), I would vote for rejection. But I would like to increase my score if the above questions could be addressed in the author response.

---

AFTER THE AUTHOR RESPONSE:

The authors' response relieved my concern about efficiency.
The proposed method is at least feasible on the online system although it brings more time cost. Thus I'd like to raise my score to above the acceptance threshold.

---

### Official Review · Reviewer_C5FP · 2022-10-26

**Confidence:** 5
**Correctness:** 4
**Technical Novelty And Significance:** 4
**Empirical Novelty And Significance:** 4
**Recommendation:** 8

**Clarity, Quality, Novelty And Reproducibility:**

The paper studies a very novel and practical problem with a simple and clean solution. The paper is well written and easy to follow.

Part of the experiment is conducted on several public dataset which makes it possible to reproduce and follow the work. However, the code is not open-sourced which limits the reproducibility;.


**Strength And Weaknesses:**

[Advantage]
1. The paper works on the user embedding generation problem which is very practical as it is widely used as the major recommendation retrieving mechanism in most internet companies.

It is known that a single embedding may not be representative enough to capture a user's multi-dimensional interest. It makes sense to use clustering based methods to generate multiple versions of embedding to increase general embedding representation power.

The research problem is practical and can benefit others.

2. The main contribution of learning cluster weight makes sense and can significantly help to improve the retrieving efficiency given the candidate quota is usually limited during the retrieval stage.  The proposed method is straightforward and technically sound.

3. The experiment section is comprehensive and the overall paper is well written and easy to follow.


[Concerns and Questions]

1.With the intra-cluster Mask M being applied to user representation learning, does it mean inter-cluster information is ignored during the embedding training?

For instance, the user's sequence for TV and Movie may be clustered into 2 different clusters, but the actions across these 2 clusters can still be beneficial to each other.

2. Since this work considers cluster based multi-interest embedding learning. It is important to analyze how the number of clusters impact the performance. To be specific, it needs to provide analysis on the special case when only using 1 cluster(i.e. does not consider multi-interest factors) vs considering multiple clusters.

3. Is online LE conducted in Pinterest production? How is the performance? The result will be more convincing with production LE/Launch.

4. Nit: In table 2, notation for vector concatenation operator should be “[;]” instead of “[:]”


**Summary Of The Paper:**

The authors work on the problem of generating multiple interest embedding to represent user interest across multiple topics via sequence modeling through user’s sequential actions. The main contribution compared to the previous papers is that the authors learns a set of weights to represent the preference over each cluster-level embedding so that the candidates can be retrieved from each interest proportionally. Experiment is conducted on an offline measure using both public dataset and private large scale dataset(Pinterest Data Set).


**Summary Of The Review:**

The paper studies a very novel and practical problem with a simple and clean solution. The paper is well written and easy to follow. The paper has the potential impact to inspire a lot of people working on the field. I recommend this paper to get accepted.

---

### Official Review · Reviewer_ufEd · 2022-10-28

**Confidence:** 4
**Correctness:** 3
**Technical Novelty And Significance:** 2
**Empirical Novelty And Significance:** 2
**Recommendation:** 3

**Clarity, Quality, Novelty And Reproducibility:**

The paper should be carefully revised to improve clarify. The contribution is limited to introducing users' multiple interests to sequential retrievers. This work can be somewhat reproducible.


**Strength And Weaknesses:**

Strengths
1. This work focuses on the encoding of user embeddings, and it is an interesting and important in recommender retrievers.
2. Incorporating time information into the multi-interest task is intriguing because user preferences change over time.
3. The authors run several experiments on various public datasets to demonstrate the effectiveness of the proposed method.

Weakness
+ The authors point that this work focuses on the retrieval in recommender systems. The models, however, are optimized using the binary cross entropy loss and tested using the metric AUC, NLL, which are rarely used in retrievers.
+ In general, the sparse and dense features can be combined with the item embedding to allow the item embeddings to be treated as a whole in models. The authors create various operations and run separate experiments on these two types of features. This means that the proposed method cannot be used in cases where there are both sparse and dense features.
+ The authors claim that "propose a multi-interest user representation model that minimizes the bias towards popular categories". However, "a higher weight is assigned to an interest cluster if the user engages more with items that belong to it". How do the authors deal with the popularity bias?
+ In Eq 9, each user chooses one interest, which is inconsistent with the goal of multi-interest.
+ The experiments are insufficiently convincing.
    + The adopted metrics AUC and NLL are more appropriate for ranking models, rather than retrievers.
    + The authors filter users with less than 100 items and thus there are only a significant limited number of users are observed. Experiments should be conducted on larger datasets. The data statistics after filtering should be more detailed. Furthermore, the authors do not provide the details about the test data.
    + Several important baseline methods in multi-interest task are missing, such as MIND [Multi-Interest Network with Dynamic Routing for Recommendation at Tmall].
    + In appendix A.1, the experiments are conducted on a synthetic dataset which is not convincing.
+ The notions, such as k in 3.1 and [;] in equations, are not well defined.


**Summary Of The Paper:**

This paper investigates the multi-interest for user embeddings in recommender retrievers. The authors consider the different weights of interests as well as time-varying interests and integrate a multi-head attention module and a cluster strategy with their weights. The proposed MIP model is then validated using publicly available datasets.


**Summary Of The Review:**

This work focuses on an important and interesting problem. However, the general solved task is not clear where the unusual objective function and metrics are adopted. There are some misleading notions and statements which makes it very confusing. The experiments are not convincing where the pre-processing is not appropriate and important baselines are missing.

---

### Decision · Program_Chairs · 2023-01-20

**Decision:**

Reject

**Justification For Why Not Higher Score:**

The submission is still not of ICLR quality.

**Justification For Why Not Lower Score:**

Described above.

**Metareview: Summary, Strengths And Weaknesses:**

This paper addresses the problem of learning multiple embeddings to represent a user's interests through the user's sequential interactions in a recommender system. The main contribution is the proposal of a new method for the task. The technique is unique in that it learns a set of weights to represent the preference over the clusters of embeddings so that the candidate items can be properly retrieved. Experiments are conducted on both public and large private datasets.

Strength
* New techniques are proposed to learn and utilize multiple embeddings for representing users' interests in a recommender system.
* The proposed method appears to be technically sound.
* Experiments are conducted on several public datasets to demonstrate the effectiveness of the proposed method.

Weakness
* The experimental results can be further improved, as indicated by the reviewers.
  * Some of the experiment settings might not be appropriate.
  * The evaluations need to be more convincing.
  * Some details of experiments are missing.
* The presentation can be improved. Many technical details can be explained better.

There are so many presentation issues pointed out by the reviewers, such as ufEd and TbF3. The authors have successfully addressed some of them. They have also presented new results. However, it is still not clear whether all the issues will be fixed if the submission is accepted.

During the rebuttal, Reviewer ufEd also checked the code and experiment settings. The reviewer is not convinced by the authors' responses. (The code and paper are available at GitHub and arxiv.)

The submission's overall quality has not reached that of ICLR. Therefore, we encourage the authors to fix all the issues and re-submit the paper.

**Summary Of Ac-Reviewer Meeting:**

I initiated a discussion.  Only one reviewer responded.

One reviewer gives reject, one accept, two marginally accept.

The experimental results are also not convincing enough. There are many presentation issues pointed out by the reviewers. The authors have fixed some of the issues.  We encourage the authors significantly improve the paper and re-submit.